# A More Globally Accurate Dimensionality Reduction Method Using Triplets

## Abstract

We first show that the commonly used dimensionality reduction (DR) methods such as t-SNE and LargeVis poorly capture the global structure of the data in the low dimensional embedding. We show this via a number of tests for the DR methods that can be easily applied by any practitioner to the dataset at hand. Surprisingly enough, t-SNE performs the best w.r.t. the commonly used measures that reward the local neighborhood accuracy such as precision-recall while having the worst performance in our tests for global structure. We then contrast the performance of these two DR method against our new method called *TriMap*. The main idea behind TriMap is to capture higher orders of structure with triplet information (instead of pairwise information used by t-SNE and LargeVis), and to minimize a robust loss function for satisfying the chosen triplets. We provide compelling experimental evidence on large natural datasets for the clear advantage of the TriMap DR results. As LargeVis, TriMap is fast and provides comparable runtime on large datasets.

## 1 Introduction

Information visualization using dimensionality reduction (DR) is a fundamental step for gaining insight about a dataset. Motivated by the fact that the humans essentially think in two or three dimensions, DR has been studied extensively (Tenenbaum et al., 2000; Kohonen, 1998; Roweis & Saul, 2000; Hinton & Roweis, 2003; Maaten & Hinton, 2008). However, how to choose the best DR method for visualizing a given dataset is yet unresolved. Despite the plethora of quantitative measures such as trustworthiness-continuity (Kaski et al., 2003), mean (smoothed) precision-recall (Venna et al., 2010), and nearest-neighbor accuracy (Van Der Maaten et al., 2009), there is no standard procedure to assess the quality of a low-dimensional embedding in general. Also these measures only focus on the local neighborhood of each point in the original space and low-dimensional embedding and fail to reflect global aspects of the data. We will argue that selecting a DR method based on these local measures is misleading.

In this paper, we mainly focus on the global properties of the data. These properties include the relative placements of the clusters in the dataset as well as revelation of outlier points that are located far away from the rest of the points in the high-dimensional space. It appears that quantifying the global properties is significantly more complicated and certainly local quality measures of DR (such as precision-recall[1]) do not capture these properties. Thus we resort to an evaluation based on visual clues that allows the practitioner to build confidence about the obtained DR results. To this end, we propose a number of transformations that happen naturally to real-world datasets. We argue that any DR method that claims to preserve the global structure of the data, should be able to handle these transformations. Ackerman et al. (2010) took a similar more theoretical approach for the task of clustering where certain natural properties are used to taxonomize different clustering algorithms.

Next, we introduce a new DR method, called TriMap. The main idea behind TriMap is to capture higher orders of structure in the data by considering the relative similarities of a chosen set of triplets of points. The method minimizes a loss over triplets that measures how well a triplet is satisfied. This loss is made robust by capping it with a damping function. We provide compelling experimental

---

[1]We verified on small datasets that other common measures such as trustworthiness-continuity (Kaski et al., 2003) are also inadequate for evaluating the global structure. Also these measures are too expensive to calculate for moderately large datasets.

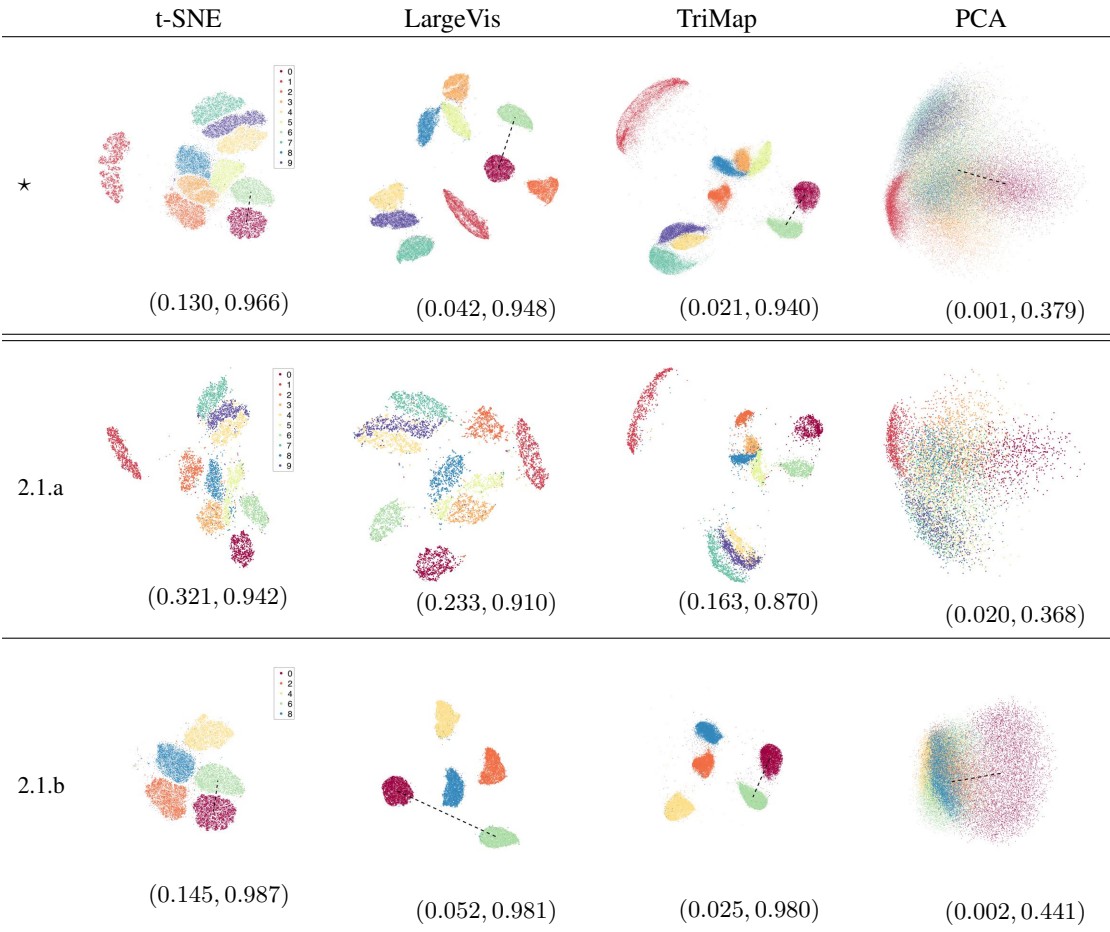

Figure 1: DR tests: ⋆) full dataset, 2.1.a) a random %10 subset, 2.1.b) even digits only. The dotted line between the centers of the clusters '0' and '6' is drawn for comparing the distance between the cluster centers before and after removing the subsets from the dataset. The (AUC, NN-Accuracy) values are shown on the bottom of each figure. Best viewed in color.

results on large natural datasets. We show that t-SNE and LargeVis, which are both based on preserving the pairwise (dis)similarities of the points, tend to form spurious clusters with clean boundaries and fail to properly reflect the global aspects of the data. On the other hand, TriMap's DR results preserve more of the global structure of the data as well as reveals outliers.

## 2 TESTING THE GLOBAL ACCURACY OF DR METHODS

Our proposed tests mimic the natural scenarios that happen in the real-world datasets. In each case we run the test on four DR methods: t-SNE, which is currently the most widely used DR method; LargeVis, a more recent method that is also based on pairwise (dis)similarities; TriMap, our new method based on triplet information; and PCA a global method which projects the data onto the two directions of largest variance. We defer the description of t-SNE & LargeVis as well as our new TriMap method to later sections. As a running example, we perform the tests on the MNIST dataset[2] which contains 70,000 images of handwritten digits (represented by 784 dimensions)[3]. We normalized the data to have pixel values between $[0, 1]$ and verified that the results are consistent over multiple runs of all methods. For each embedding, we calculated the area under the mean precision-

---

[2]http://yann.lecun.com/exdb/mnist/
[3]For ease of comparison, we use the same initial solution for t-SNE and TriMap, whenever possible. The current implementation of LargeVis does not support initial solutions.

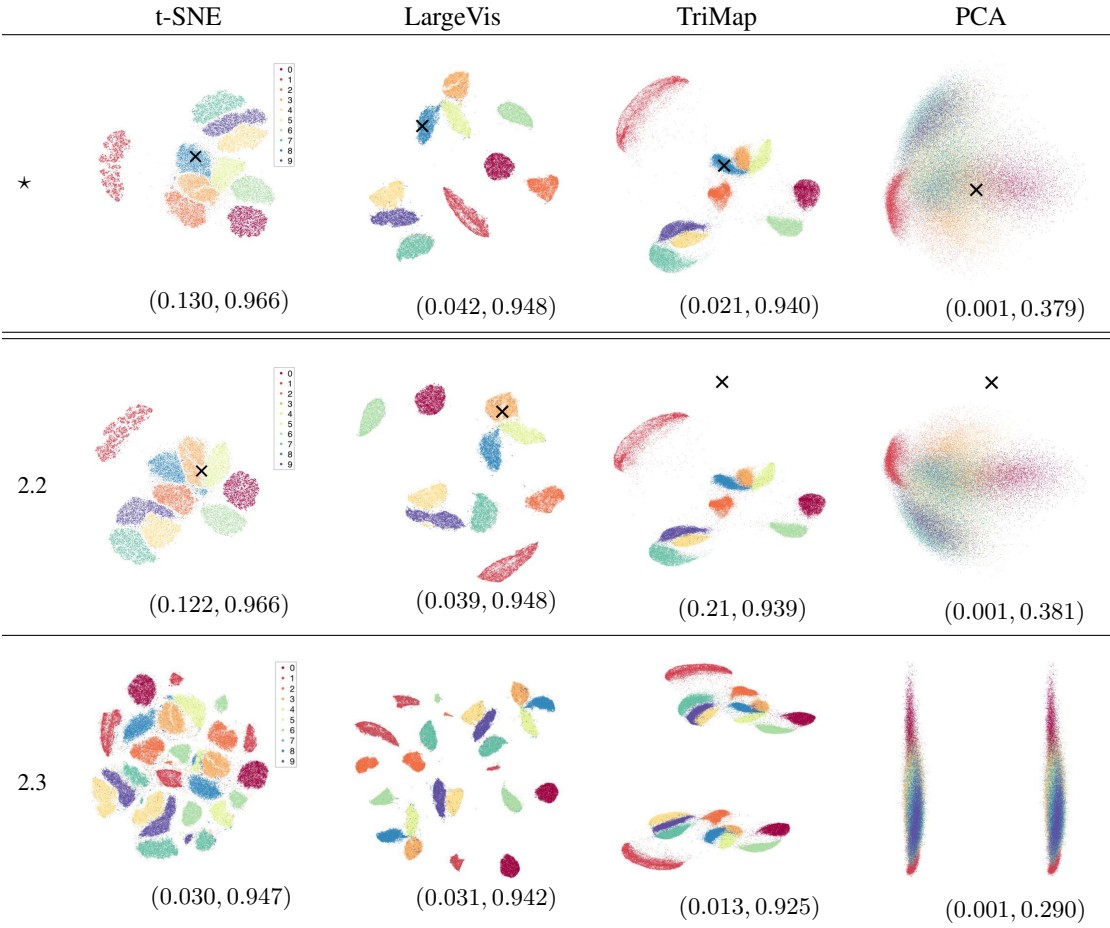

Figure 1: (continued) DR tests: ⋆) full dataset, 2.2) outlier, 2.3) multiple scales. The (AUC, NN-Accuracy) values are shown on the bottom of each figure. Best viewed in color.

recall curve (AUC) (Venna et al., 2010).[4] We also calculate the nearest-neighbor classification accuracy for each embedding. The (AUC, NN-Accuracy) values are shown on the bottom of each figure. The main reason for including these local measure in the results is to emphasize the fact that these measures fail to reflect the globally properties of the embedding, discussed in the following. Figure 5(⋆) shows the results on the full dataset by the four methods.

**Partial observation test**   A DR tool should be invariant to removing a subset of points from the dataset. The placement of the remaining points (or clusters) after running the DR method on the reduced data set should remain relatively unchanged compared to the embedding created from the full dataset. In a first test, the subset of removed points are selected at random. This test mimics the fact that in reality we only see a sample from the data distribution and the DR method should be stable w.r.t. the sample size. Figure 5(2.1.a) shows the results after removing %90 of the dataset at random. t-SNE and TriMap both produce good results in that the clusters of the reduced datasets are located roughly in the same arrangement as in the full dataset. However, LargeVis moves the clusters around (e.g. cluster '1').

For labeled datasets, we might be interested in visualizing a particular subset of the classes, leading to our second test. Figure 5(2.1.b) gives the results on running the methods on just the subset of even digits of MNIST. As can be seen, only the TriMap method is able to preserve the relative distances between all the clusters after removing the odd digits, while t-SNE and LargeVis place the remaining clusters at arbitrary positions.

---

[4]More precisely, for each point we fix a neighborhood size of 20 in the high-dimension as the "relevant points" and vary the neighborhood size in the low-dimension between 1 and 100. This gives us a precision and recall curve for each point. The AUC measure is the area under the mean curve for all points.

**Outlier test**   Natural datasets frequently contain outliers due to measurement or experimental errors. Detecting and removing the outliers is key step in statistical analysis that should be facilitated with the DR methods. We can check the ability of the DR methods to reveal outliers by adding artificial outlier points to the original dataset. Figure 5(2.2) shows the DR results after shifting point **x** (originally placed inside cluster '8' by all methods) far away in a random direction. The point **x** is clearly an outlier in the new dataset (as verified by the PCA result). Surprisingly, t-SNE and LargeVis both place the outlier inside the cluster '3', which happens to be the nearest cluster to this point in higher dimension. More disturbingly, adding a single outlier this way consistently rearranges the relative location of the clusters for t-SNE and LargeVis. TriMap shows the outlier and preserve the structure.

**Multiple scales test**   A DR tool should be able to reflect the global structure of the data at different scales. For instance, a dataset might consist of multiple clusters where each cluster itself may contain multiple sub-clusters and so on. The practitioner can do a rudimentary test by duplicating and shifting the natural dataset at hand. That is, for each point $x_n$ of the original dataset, we add a point $x_n + c$, where $c$ is a fixed random shift which is chosen large enough such that the two copies are guaranteed to be far apart in the high-dimensional space[5]. When applied on the duplicated dataset, the DR method should be able to show each copy separately. Figure 5(2.3) illustrates the results on a duplicated MNIST dataset. We expect to see two identical copies of the same dataset in the low-dimensional embedding, which can be verified by the PCA result. Curiously enough, both t-SNE and LargeVis fail to provide any meaningful global structure[6]. In fact, both methods tend to split some of the clusters into smaller sub-clusters. On the other hand, TriMap successfully recovers the two identical copies, which look similar to the original dataset in Figure 5(⋆).

## 3   SKETCH OF THE t-SNE AND LARGEVIS

The t-SNE method (Maaten & Hinton, 2008) is perhaps the most commonly used DR method in practice. The main idea of t-SNE is to reflect the pairwise similarities of the points by minimizing a KL divergence between the normalized pairwise similarities in the high-dimensional space and the low-dimensional embedding. The t-SNE method has $\mathcal{O}(N^2)$ time complexity for $N$ points. However, the complexity can be reduced to $O(N \log N)$ by approximating the gradient using tree-based algorithms (Van Der Maaten, 2014). t-SNE is easy to apply to general datasets and usually produces nicely separated clusters. Also it has been proven theoretically (Arora et al., 2018) that if the clusters are separated far enough in higher dimension, then $t$-SNE separates them in the embedding[7]. However, as we showed in the previous section, the DR results produced by $t$-SNE are sometimes extremely misleading: the whole dataset is simply collapsed into an "orb" and the outliers are shown as "inliers". The latter can be explained by fact that for an outlier point, the pairwise similarities to the points that are closest to the outlier point dominate the rest, causing the method to pull back the outlier point to the closest cluster.

LargeVis is a more recent method (Tang et al., 2016) that also aims to preserve the *pairwise* (dis)similarities of the points in the high-dimensional space in the low-dimensional embedding. To avoid the $\mathcal{O}(N^2)$ complexity of t-SNE, LargeVis uses a negative sampling approach (Mikolov et al., 2013) to randomly select a subset of the dissimilar pairs. However, the weights of all the dissimilar pairs are set to a positive constant. This causes the LargeVis to lose global information in the data, as we showed in the previous section. Overall, LargeVis forms well-separated clusters. However, the outlier points that are far away or in-between the clusters are pushed back into the closest clusters. This is a consequence of using a noisy distribution for negative sampling which tends to pick points from the denser regions with higher probability. For example, moving single points that lie between multiple large clusters inside the nearest cluster increases the likelihood of the model (because a single point has a very small probability of being selected as a dissimilar example by any of points that lie inside the large clusters).

---

[5]Similar results obtained by shifting the datasets in 4 directions.

[6]Verified with a large range of the perplexity parameter for t-SNE (from the default of 30 to 150).

[7]The proof requires that "early exaggeration" is used in all iterations of training.

## 4 THE NEW TRIMAP METHOD

The main idea in developing the TriMap method is to preserve a higher-order of structure in the data. In other words, we aim to reflect the relative (dis)similarities of triplets of points, rather than pairs of points. Formally, a triplet is a constraint between three points $i$, $j$, and $k$, denoted as an ordered tuple $(i, j, k)$, which indicates: "point $i$ is more similar to point $j$ than point $k$".

**Problem formulation** Let $\{x_n\}_{n=1}^N$ denote the high-dimensional representation of $N$ points. Our goal is to find a lower-dimensional representation $\{y_n\}_{n=1}^N$ for these points (in 2D or 3D). Let $\tilde{p}_{ij} \geq 0$ be a pairwise similarity function between $x_i$ and $x_j$ in high dimension and $\tilde{q}_{ij} \geq 0$ a similarity function between $y_i$ and $y_j$ in low dimension. We denote by $T$ be the set of all triplets $(i, j, k)$ where $\tilde{p}_{ij} > \tilde{p}_{ik}$, i.e., $T = \{(i, j, k) : \tilde{p}_{ij} > \tilde{p}_{ik}\}$. A low-dimensional embedding can be calculated by minimizing the following objective

$$\min_{\{y_n\}} \sum_{(i,j,k)\in T} \omega_{ijk} \frac{\tilde{q}_{ik}}{\tilde{q}_{ij}} \tag{1}$$

where $\omega_{ijk}$ is the weight that reflects the importance of the triplet $(i, j, k)$ and the ratio $\frac{\tilde{q}_{ik}}{\tilde{q}_{ij}} \geq 0$ can be seen as the loss associated with the triplet $(i, j, k)$. Note that as $\tilde{q}_{ik}$ becomes smaller and $\tilde{q}_{ij}$ becomes larger, the loss approaches zero.

Typically not all the triplets can be satisfied when mapping from high-dimension to lower-dimension, i.e., we cannot have $\tilde{q}_{ij} > \tilde{q}_{ik}$, for all $(i, j, k) \in T$. The issue arises because of the lower degree of freedom in the low-dimensional space. As a simple example, consider mapping uniformly distributed points on a two-dimensional circle to a one-dimensional line; regardless of the choice of embedding, the end points of line will always violate some triplet constraints. For a highly violated triplet $(i, j, k)$ with $\tilde{q}_{ik} \gg \tilde{q}_{ij}$, the triplet loss $\frac{\tilde{q}_{ik}}{\tilde{q}_{ij}}$ will become too large and dominate the loss minimization over all triplets. Thus damping the effect of individual triplets is crucial whenever the triplets are sampled from a high-dimensional space. We use properties of the generalized $\log$ and $\exp$ functions to define robust versions of the weighted triplet loss:

We propose the following objective

$$\min_{\{y_n\}} \sum_{(i,j,k)\in T} \omega_{ijk} \log_t \left(1 + \frac{\tilde{q}_{ik}^{(t')}}{\tilde{q}_{ij}^{(t')}}\right), \tag{2}$$

where the $\log_t$ is defined as (Naudts, 2002):

$$\log_t(x) = \begin{cases} \log(x) & \text{if } t = 1 \\ (x^{1-t} - 1)/(1-t) & \text{otherwise.} \end{cases}$$

Figure 2: (left) Loss transformation $\ell \to \log_t(1 + \ell)$ and (right) similarity function $u \to \exp_{t'}(-u^2)$ for different values of $t$ and $t'$, respectively.

Note that $\log_t$ is concave and non-decreasing and generalizes the $\log$ function which is recovered at $t = 1$. Thus Equation (2) includes the log-transformation as a special case (see Figure 2(left, red)). However, for $x > 1$, the $\log_t$ function with $t > 1$ grows slower than the $\log$ function and reaches the value $1/(t-1)$ in the limit $x \to \infty$. Therefore, for $t > 1$, each triplet can contribute at most $1/(t-1)$ to the total loss and the objective function of TriMap in Equation (2) is robust to violated triplets.

Also inspired by the good properties of the heavy-tailed distributions for low-dimensional embedding (Maaten & Hinton, 2008; Yang et al., 2009), we define a parameterized similarity function

$$\tilde{q}_{ij}^{(t')} = \exp_{t'}(-\|y_i - y_j\|^2), \quad \text{where } \exp_{t'}(x) = \begin{cases} \exp(x) & \text{if } t' = 1 \\ [1 + (1-t')x]_+^{1/(1-t')} & \text{otherwise.} \end{cases} \tag{3}$$

Here $[\cdot]_+ := \max(0, \cdot)$. Notice that the tail-heaviness of $\tilde{q}_{ij}^{(t')}$ increases with the value of $t'$. That is, the Gaussian function is recovered at $t' = 1$ and $t' = 2$ amounts to the Student t-distribution with one degree of freedom. For $t' > 2$, we recover distributions with even heavier tails. Figure 2(right) illustrates the $\tilde{q}_{ij}^{(t')}$ function for several values of $t'$. In all our experiments, the default parameters for TriMap were set to $t = t' = 2$, but we will briefly explore the other choices in Section 4.

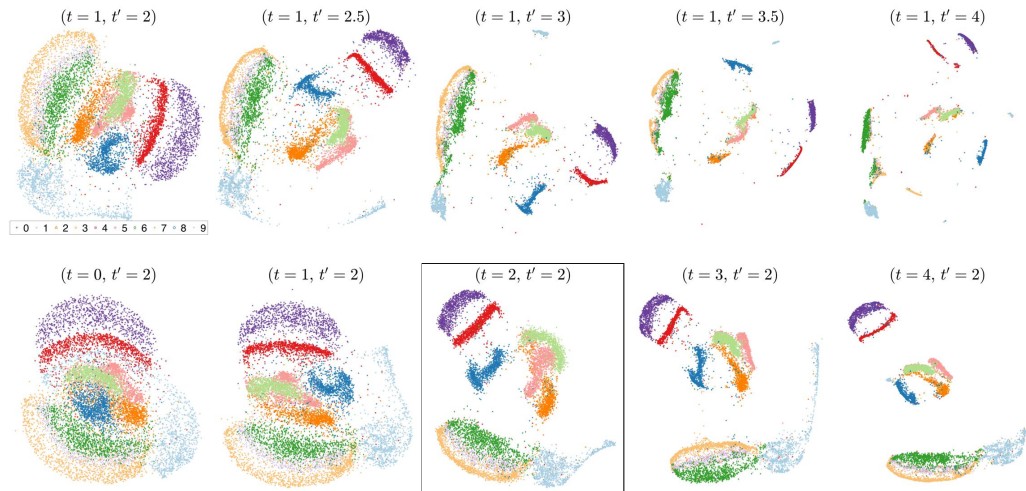

Figure 3: Effect of changing $t$ and $t'$ parameters: top) results using fixed $\log$-transformation ($t = 1$) and different values of $t'$ for similarity function. Notice that having heavier-tail provides more separability but cannot fix the clutter introduced by the unsatisfied triplets by itself, bottom) results using different values of $t$ (i.e. $\log_t$-loss transformation) with $t' = 2$ (i.e. Student t for similarities) fixed. Note that having $\log_t$-transformation is crucial to obtain nice visualizations. Value of $t = 0$ corresponds to no transformation (Equation (1)), $t = 1$ recovers the $\log$-transformation, $t = 2$ is used in our experiments. The box indicate the default choice of parameters by our algorithm.

Another crucial component of TriMap is the weight $\omega_{ijk}$ assigned to the triplet $(i, j, k)$, quantifying the importance for satisfying $(i, j, k)$ (See Appendix C for an experimental justification):

$$\omega_{ijk} := \frac{\tilde{p}_{ij}}{\tilde{p}_{ik}}, \quad \text{where } \tilde{p}_{ij} = \exp(-\frac{\|x_i - x_j\|^2}{\sigma_i \sigma_j}). \tag{4}$$

The scale factor $\sigma_i$ is set to the average distance of $x_i$ to its 10-th, 11-th, ... 20-th nearest-neighbors. This choice of $\sigma_i$ adjusts the scale of $\tilde{p}_{ij}$ to the density of the data (Zelnik-Manor & Perona, 2005).

**Triplet sampling** For $N$ data points, the number of triplets is $\mathcal{O}(N^3)$. Thus the minimization of (2) with all the possible is too expensive. However, in practice, there exists a large amount of redundancy among the triplets. For instance, the two triplets $(i, j, k)$ and $(i, j, k')$, in which $k$ and $k'$ are located close together, convey the same information. We now develop a heuristic for sampling an $O(N)$ size subset of triplets, such that the mapping produced from the subset essentially reproduces the mapping derived from all triplets (evidence not shown for lack of space). We sample two types of triplets to form the embedding.

**Nearest-neighbors triplets:** For each point $i$, the closer point $j$ is selected from $m$-nearest neighbors of the point and the distant points are sampled uniformly from the subset of points that are farther away from $i$ than $j$. Note that this choice of triplets reflect the local as well as the global information of the data. For each point in the set of $m$-nearest neighbors, we sample $m'$ distant points which results in $N \times m \times m'$ triplets in total. In all our experiments, we use $m = 50$ and $m' = 10$.

**Random triplets:** These triplets are sampled uniformly at random from the set of full triplet set $T$. Random triplets help preserve the global structure of the data. We add $s$ randomly selected triplets for each points, which results in $N \times s$ triplets in total. In all our experiments, we set $s = 5$.

The overall complexity of the algorithm is bounded by the nearest neighbor search step, which for e.g. Nearest-Neighbor-Descent (Dong et al., 2011) has been shown empirically to have $\mathcal{O}(N^{1.14})$ complexity. However, after the nearest-neighbor calculation step, we achieve linear complexity. In practice, dividing each ratio $\frac{\tilde{p}_{ij}}{\tilde{p}_{ik}}$ at the end by the maximum ratio among the triplets in the set of sampled triplets and adding a constant positive bias $\gamma > 0$ to all ratios improves the results. In all our experiments, we set $\gamma = 0.001$.

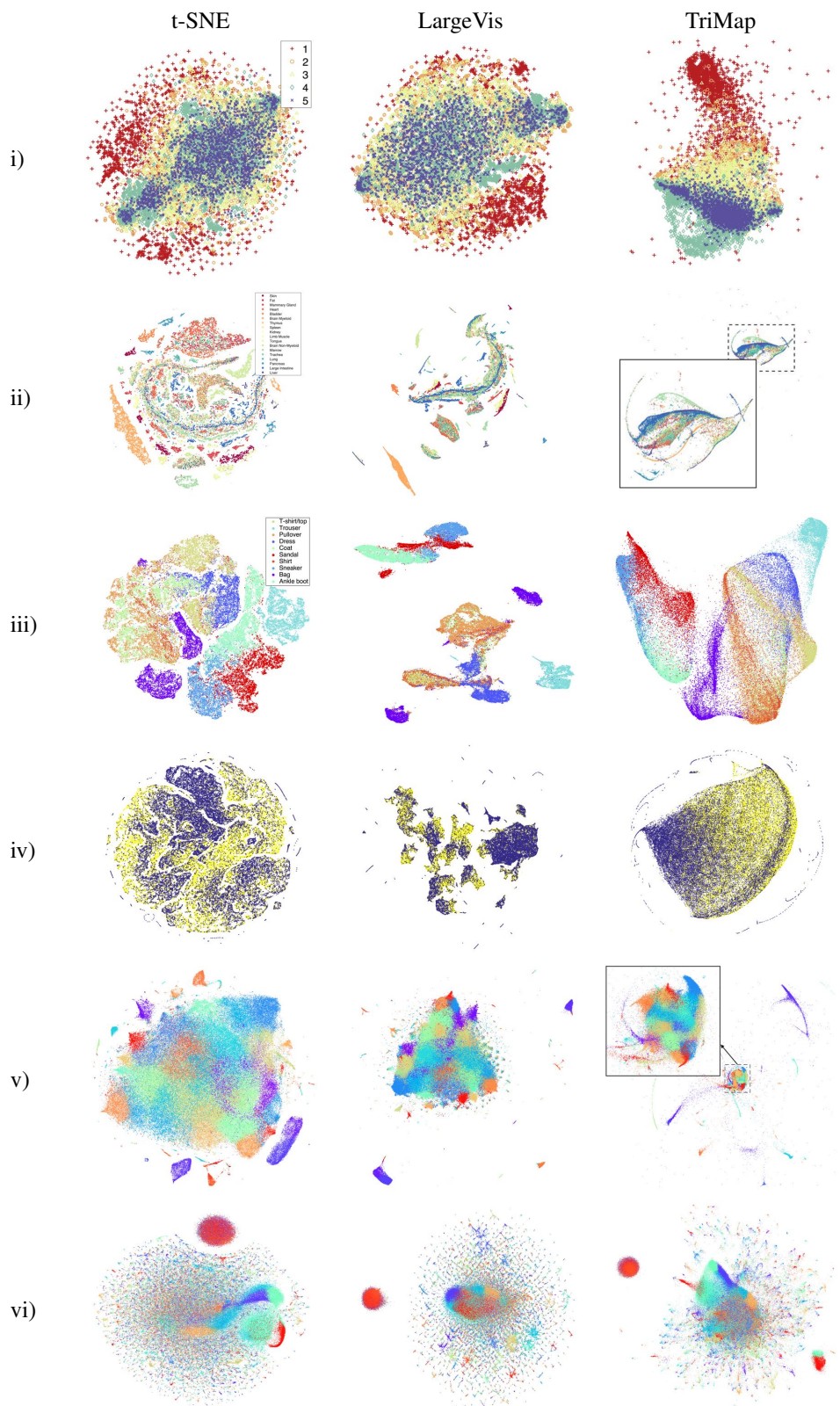

Figure 4: Results of different methods: i) Epileptic Seizure Recognition, ii) Tabula Muris, iii) Fashion MNIST dataset, iv) TV News: commercial (yellow), non-commercial (blue), v) 380k+ lyrics, vi) DBLP dataset: collaboration network of authors. Best viewed in colors.

**Effect of changing $t$ and $t'$** We show the effect of changing the parameters $t$ and $t'$ in Figure 3. The experiments are performed on the USPS Digits dataset [8]. In the first set of experiments, we fix $t = 1$ (i.e. use $\log$-transformation) and increase the value of $t' = 2$ (i.e. the tail-heaviness of the similarity function). It is clear from the results that having heavier-tail provides more separability but cannot fix the clutter introduced by the unsatisfied triplets. Next, we fix the similarity function to Student t-distribution ($t' = 2$) and increase the value of $t$ for the $\log_t$-transformation. Value of $t = 0$ corresponds to no transformation (Equation (1)) and $t = 1$ recovers the $\log$-transformation. Note that larger values of $t$ provides denser clusters and more separate clusters and reduces the clutter introduced by unsatisfied triplets. We use $t = t' = 2$ in our experiments.

## 5 EXPERIMENTS

We evaluate the performance of TriMap for DR on a number of real-world datasets and compare the results to t-SNE and LargeVis[9]. We also compared our results to triplet embedding methods such as STE (van der Maaten & Weinberger, 2012). However, the results were clearly inferior and are omitted due to space constraints. The code for TriMap as well as the datasets for the experiments are available online[10]. In all our experiments, we use the default values $t = t' = 2$. All experiments are performed on a single machine with 2.6 GHz Intel Core i5 processor and 16 GB RAM. We provide DR results on six datasets:

i) Epileptic Seizure Recognition[11]: contains 11,500 EEG recordings of patients under 5 different conditions. Each data point is one second of measurements with 178 samples / dimensions.

ii) Tabula Muris[12]: contains 53,760 samples from 18 different mouse tissues in 23,433 dimensions.

iii) Fashion MNIST[13]: consists of 70,000 examples. Each example is a $28 \times 28$ gray scale image (784 dimensions) from 10 classes of clothing items.

iv) TV News[14]: contains audio-visual features extracted from 150 hours of TV broadcast from 5 different TV channels. The dataset contains 129,685 instances, labeled as commercial/non-commercial, and 4120 features.

v) 380k+ lyrics[15]: the dataset contains lyrics of 380k+ different songs, gathered from `http://www.metrolyrics.com`. We first form a weighted co-occurrence graph for the words based on the frequency of the pairs of words which occur together in a sliding window of length 5 words. Next, we compute the representation of each word using the LINE method Tang et al. (2015). We use number of negative samples equal to 5 and a learning rate of 0.025 with number of training samples equal to 1000M. We map each word to a 256 dimensional vector and then, calculate the representation of each song by a weighted average of the words that occur in the song. After removing the stop words and the songs with no available lyrics, we get 266,557 instances.

vi) DBLP collaboration network[16]: the collaboration network of 317,080 authors. Two authors are connected in the network if they publish at least one paper together. We use LINE to find a 256 dimensional representation of the network before applying the methods. We set number of negative samples to 5, the learning rate to 0.025, and the number of training samples to 3000M.

In our experiments, we use the `sklearn` implementation of fast t-SNE with the default parameters and learning rates. For LargeVis, we use the official implementation[17] with the default setting of the learning equal to 1.0 and number of negative samples equal to 5. For TriMap, we reduce the

---

[8]`https://cs.nyu.edu/~roweis/data.html`

[9]Recently, a new DR method called UMAP was proposed by McInnes & Healy (2018). Although the running time of UMAP is comparable with TriMap, we did not include the UMAP results for two reasons: 1) the method is only described in an arXiv paper, 2) despite the different motivation, the results are strikingly similar to the LargeVis reported here.

[10]`https://github.com/ANONYMOUS/trimap`

[11]`https://archive.ics.uci.edu/ml/datasets/Epileptic+Seizure+Recognition`

[12]`https://github.com/czi-hca-comp-tools/easy-data/blob/master/datasets/tabula_muris.md`

[13]`https://github.com/zalandoresearch/fashion-mnist`

[14]`http://archive.ics.uci.edu/ml/datasets/tv+news+channel+commercial+detection+dataset`

[15]`https://www.kaggle.com/gyani95/380000-lyrics-from-metrolyrics/data`

[16]`https://snap.stanford.edu/data/com-DBLP.html`

[17]`https://github.com/lferry007/LargeVis`

| Dataset (size, dimension) | t-SNE | LargeVis | TriMap |
|---|---|---|---|
| Epileptic Seizure (11,500, 178) | 0-00:07:34 | 0-00:06:55 | **0-00:03:13** |
| Tabula Muris (53,760, 23,433) | 1-11:25:22 | 0-1:59:53 | **0-00:15:14** |
| Fashion MNIST (70,000, 784) | 0-00:26:33 | **0-00:15:15** | 0-00:19:44 |
| TV News (129,685, 4120) | 1-23:12:49 | 0-02:04:04 | **0-00:37:11** |
| 380k lyrics (266,557, 256) | 0-17:12:40 | **0-01:03:02** | 0-02:12:38 |
| DBLP (317,080,256) | 1-02:46:08 | **0-01:18:15** | 0-02:38:07 |

Table 1: Runtime of the methods in `d-hh:mm:ss` format.

dimensionality of data to 100 if necessary, using PCA and apply ANNOY[18] (a random projection based approximate nearest neighbor search method) to calculate the nearest neighbors. However, in some cases the random projection based nearest neighbor search fails to correctly estimate the nearest neighbors of outlier points. This can be resolved by finding a small number (typically 5) of nearest neighbors using the Balltree algorithm and combining the results with ANNOY. Further details and hints are provided in our implementation available online. We use $m = 50$, $m' = 5$, and $s = 5$ for all datasets. The optimization is done with batch gradient descent and adaptive learning rate.

Figure 4-i) illustrates the results on the Epileptic Seizure Recognition dataset. TriMap shows a smooth transition between measurements under different conditions while both t-SNE and LargeVis fail to preserve the global structure. Figure 4-ii) shows the results on the Tabula Muris dataset. As can be seen, TriMap reveals multiple outlier points which are not recovered using t-SNE and LargeVis. These outlier points can be verified by considering the ratio of the average distance of the top $5\%$ farthest points from the center of the dataset to the average distance to the center of the remaining points. The actual ratio in high-dimensions equals to $5.32$ where in low-dimension, equals to $1.50$, $1.02$, and $1.95$ for t-SNE, LargeVis, and TriMap, respectively. The results on the Fashion MNIST dataset are shown in Figure 4-iii). t-SNE and LargeVis tend to create many spurious clusters. For instance, the cluster of "Bags" is divided into multiple smaller sub-clusters. On the other hand, TriMap represents a smooth embedding where each class in concentrated around the same region and there appears to be a smooth transitions among different classes. Figure 4-iv) shows the results on the TV News dataset. Bothe t-SNE and LargeVis tend to produce several patches of points that do not have a clear structure. TriMap on the other hand shows a smooth transition between the two classes with a few outliers that are present around the main cluster. Figure 4-v) shows the results on the 380k+ lyrics dataset. We use the 50 clusters in the high-dimensional space found by the k-means++ algorithm as labels. Unlike the other two methods, TriMap reveals multiple layers of structure with several outliers in the dataset. The main cluster that is shown both by t-SNE and LargeVis is also shown in the center of the map with TriMap. Finally, Figure 4-vi) shows the results on the DBLP dataset. As labels, we use the 50 clusters in the high-dimensional space found by the k-means++ algorithm. The results of LargeVis and TriMap are comparable. However, LargeVis tends to create more small clusters by grouping the outlier points that are close together.

Finally, the runtime of the methods is shown in Table 5. TriMap is consistently faster than t-SNE and in some cases, outperforms LargeVis in terms of runtime. The main advantage of TriMap is evident on dataset with very large number of dimensions.

## 6 CONCLUSION

Dimensionality reduction will remain a core part of Machine Learning because humans need to visualize their data in 2 or 3 dimensions. We formulated a number of transformations and outlier tests that any practitioner can use to test the visualization method at hand. The main existing methods dramatically fail our tests. We also present a new DR method that performs significantly better. This new method called TriMap is based on triplet embedding but otherwise incorporates a number of redeeming features employed by the predecessors such as the use of the heavy tailed distributions and sub sampling for obtaining linear time complexity (after the nearest-neighbor calculation step). Our results on wide range of datasets are quite compelling and suggest that further advances can be made in this area. Our next goal is to apply TriMap to data domains where we can test the quality of the obtained visualization and outlier detection by further physical experiments.

---

[18] https://github.com/spotify/annoy

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

## A   RELATION TO T-STOCHASTIC TRIPLET EMBEDDING

One of the commonly used methods to find low-dimensional representation of a set of items using only triplet information is the t-Stochastic Triplet Embedding method (t-STE) (van der Maaten & Weinberger, 2012). We show here that t-STE is a special case of our method when the temperatures defining our loss is chosen non-optimally and no weighting of the triplets is used.

Given a set of triplets $T = \{(i, j, k)\}$ over $N$ items, the goal of t-STE is to find an embedding $\{y_n\}_{n=1}^N$ in the low-dimensional space such that the following relation holds with high probability:

$$\|y_i - y_j\| \leq \|y_i - y_k\|, \quad \text{for each } (i, j, k) \in T.$$

Let $\tilde{q}_{ij}^{\text{STE}} \geq 0$ denote the similarity function between $y_i$ and $y_j$. In t-STE, the similarity function is chosen to be a Student t-distributions with $\alpha$ degrees of freedom, that is

$$\tilde{q}_{ij}^{\text{STE}} = \left(1 + \frac{\|y_i - y_j\|^2}{\alpha}\right)^{-\frac{1+\alpha}{2}}.$$

Note that this similarity function is a special case[19] of our similarity function $\tilde{q}_{ij}^{(t')}$ (defined in Equation (**??**) with temperature $t' = \frac{3+\alpha}{1+\alpha}$, where all the instances are scaled by a constant[20] factor $\frac{1+\alpha}{2\alpha}$. The *satisfaction probability* of the triplet $(i, j, k)$ is defined as

$$\text{Pr}_{ijk} = \frac{\tilde{q}_{ij}^{\text{STE}}}{\tilde{q}_{ij}^{\text{STE}} + \tilde{q}_{ik}^{\text{STE}}} = \frac{1}{1 + \frac{\tilde{q}_{ik}^{\text{STE}}}{\tilde{q}_{ij}^{\text{STE}}}}.$$

Notice that $\text{Pr}_{ijk} \to 1$ whenever $\tilde{q}_{ij}^{\text{STE}} \gg q_{ik}^{\text{STE}}$. In t-STE, the low-dimensional embedding is computed by minimizing the sum of negative log-probabilities over all triplets

$$\min_{\{y_n\}} -\sum_{(i,j,k)\in T} \log \text{Pr}_{ijk} = -\sum_{(i,j,k)\in T} \log \frac{1}{1 + \frac{\tilde{q}_{ik}^{\text{STE}}}{\tilde{q}_{ij}^{\text{STE}}}} = +\sum_{(i,j,k)\in T} \log_1\left(1 + \frac{\tilde{q}_{ik}^{\text{STE}}}{\tilde{q}_{ij}^{\text{STE}}}\right).$$

The above rewrite of t-STE's objective shows that it is a special case of our loss (2) with temperature $t = 1$ and when all triplet weights $\omega_{ijk}$ are set to one. As we discussed in Section 4, the ratio $\frac{\tilde{q}_{ik}^{\text{STE}}}{\tilde{q}_{ij}^{\text{STE}}}$ may be seen as the loss of the triplet $(i, j, k)$. The transformation $\log(1 + \frac{\tilde{q}_{ik}^{\text{STE}}}{\tilde{q}_{ij}^{\text{STE}}})$ used in the t-STE objective reduces the effect of each triplet[21]. However for $t = 1$, the effect of each individual triplet is still unbounded and the method becomes too outlier dependent. This can be verified by the results in Figure 3 where t-STE (with additional triplet weights) corresponds to $(t = 1, t' = 2)$ whereas our default temperature choice is $(t = 2, t' = 2)$. Note that by choosing $t > 1$, the loss of each individual triplet becomes bounded by $1/(t - 1)$, and therefore the total loss becomes more robust to unsatisfied triplets.

## B   DR TESTS FOR PHATE, UMAP, AND T-STE

In this section we provide the results of the DR tests introduced in Section 2 for the following additional methods: PHATE (Moon et al., 2017), UMAP (McInnes & Healy, 2018), and t-STE (van der Maaten & Weinberger, 2012). Note that t-STE was originally used for finding a low-dimensional embedding for a set of item based on a set of triplet comparisons when the high-dimensional representation was unknown. The results on the DR tests are given in Figure 5. Note that based on the discussion in Appendix A, t-STE corresponds to the special case of TriMap where $(t = 1, t' = 2)$ and all weights are set to 1.0.

The results on the original dataset are shown in Figure 5.$\star$. Note that t-STE shows many factitious outliers in between, or even away from the clusters. This phenomenon becomes worse as the size of

---

[19]Vanilla STE with Gaussian similarities is also a special case where $t' = 1$.

[20]For $\alpha = 1$, this corresponds to $t' = 2$ (our default) and the scaling disappears.

[21]The $\log(1 + \cdot)$ transformation has been used in the context of ranking to reduce the effect of individual losses on the overall ordering Yun et al. (2014).

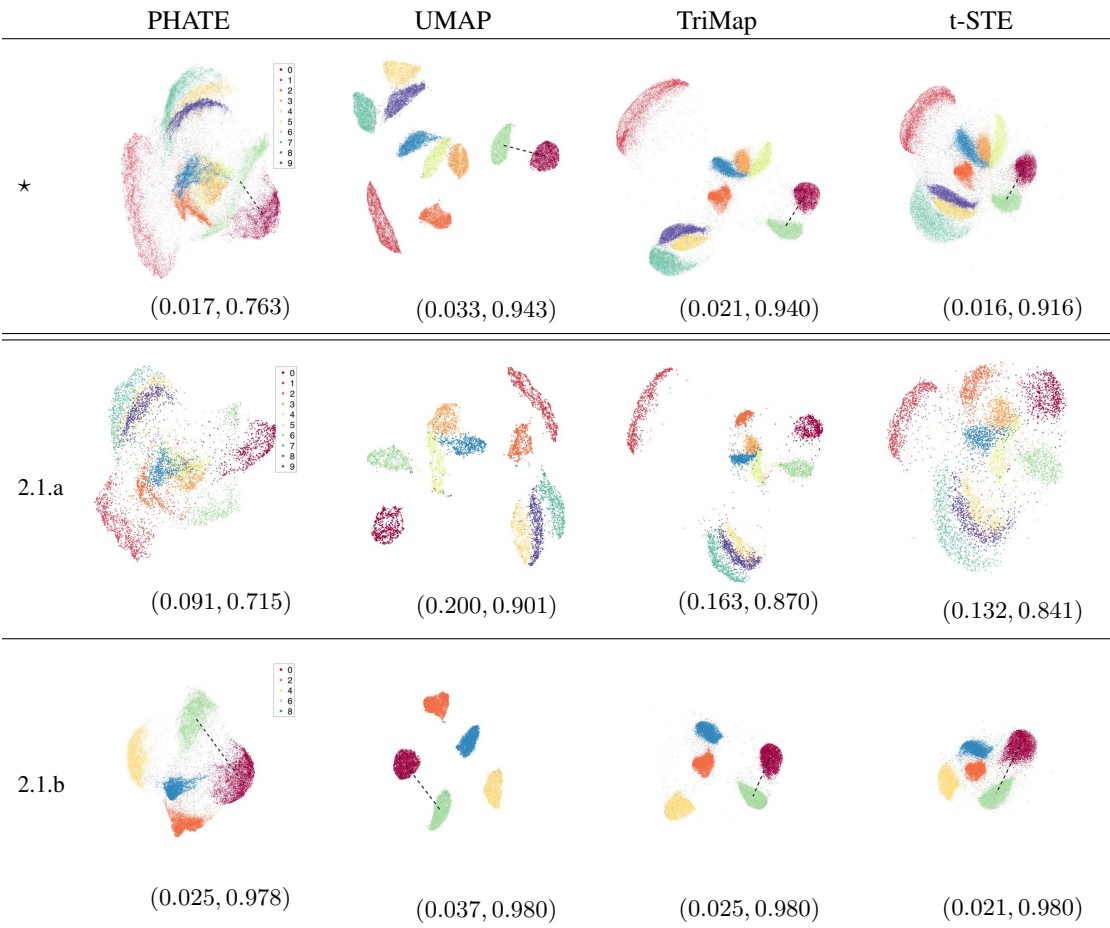

Figure 5: DR tests: ⋆) full dataset, 2.1.a) a random %10 subset, 2.1.b) even digits only. The dotted line between the centers of the clusters '0' and '6' is drawn for comparing the distance between the cluster centers before and after removing the subsets from the dataset. The (AUC, NN-Accuracy) values are shown on the bottom of each figure. Best viewed in color.

the dataset is decreased (Figure 5.2.1.a). UMAP performs well on the $10\%$ subset, however, PHATE splits the cluster of 6's. On the subset of even digits, all the methods perform well. However, t-STE again introduces many factitious outliers. Figure 5.2.2 shows the effect of adding an outlier point (marked with **X**). Except for TriMap, all the other methods fail to reveal the outlier. In fact, all other methods pull the outlier into (the closest) cluster. Finally, the result of copying and shifting the dataset is shown in Figure 5.2.3. PHATE, TriMap, and t-SNE successfully display the two copies. However, the quality of each copy deteriorates in the embedding created by the PHATE method. Also, t-STE again introduces many factitious outliers. Finally, the result for UMAP is similar to the LargeVis method; the clusters of each copy is preserved, however, the relative distances between the clusters are totally lost.

In summary, we are not aware of any DR method that is competite with TriMap on our tests for global accuracy. Also the (AUC,NN-Accuracy) numbers reported in Figure 5 are not helpful.

## C   THE EFFECT OF WEIGHTS, TRIPLET TYPE AND NUMBER OF TRIPLETS

Recall that TriMap picks $m \times m'$ triplets per point in the dataset, where $m$ is the number of nearest neighbors and $m'$ are distant points. We also add $s$ random triplets per data point. In this section we show the effect of changing these parameters as well as the effect of weighting the triplets (as defined

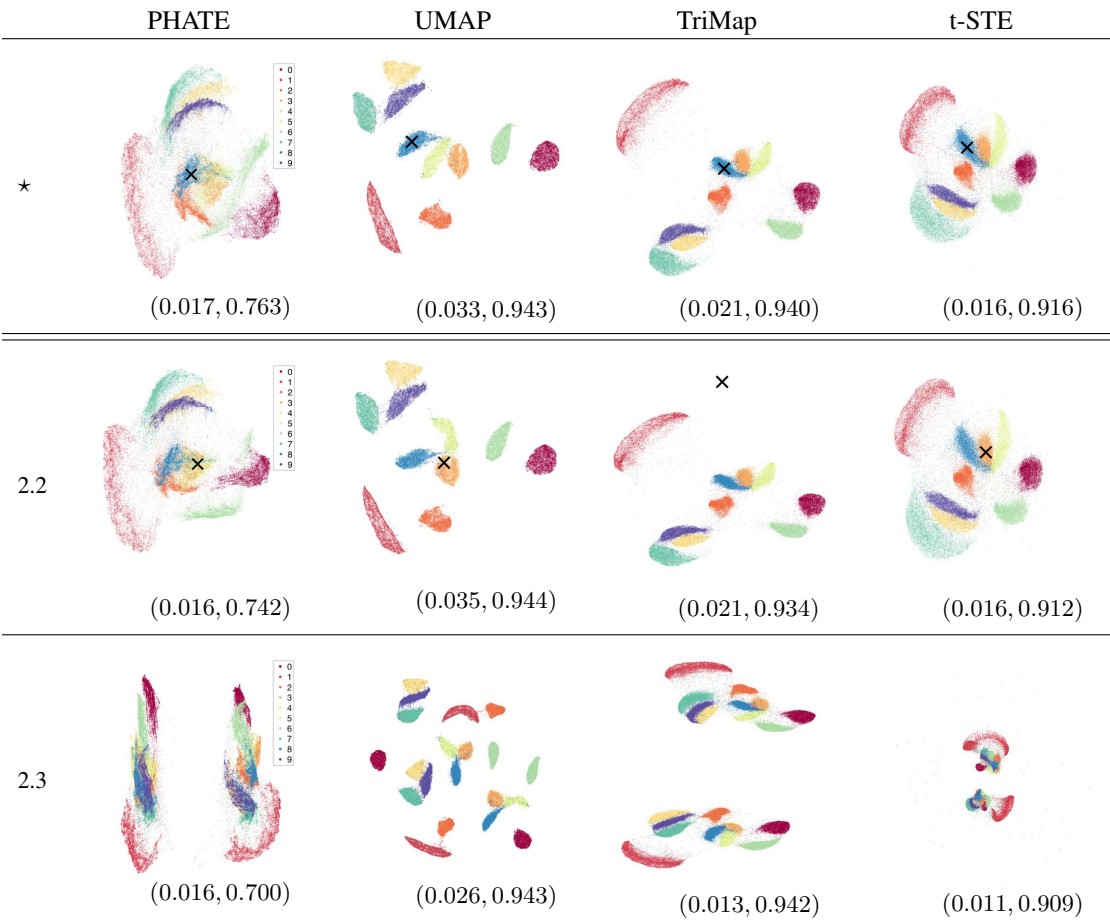

Figure 5: (continued) DR tests: ⋆) full dataset, 2.2) outlier, 2.3) multiple scales. The (AUC, NN-Accuracy) values are shown on the bottom of each figure. Best viewed in color.

in (4)). Recall that TriMap uses these weights and default values $m = 50$, $m' = 10$ and $s = 5$. We conduct the experiments on the USPS dataset.

Figure 6 shows the effect of changing the number of nearest-neighbors $m$ while fixing the number of distant points $m' = 10$ and number of random triplets per point $s = 0$. We show the results with and without triplet weighting, i.e. for experiments with unweighted triplets, we set $\omega_{(i,j,k)} = 1$ for all $(i, j, k) \in T$. As can be seen, for weighted triplets, increasing the number of nearest neighbors consistently improves the results. However, using unweighted triplets introduces multiple factitious outliers. Additionally, when using unweighted triplets, increasing $m$ does not improve the results as much.

Figure 7 shows the results where number of nearest neighbors $m = 10$ and random triplets per point $s = 0$ are fixed and the number of distant points $m'$ is varied. Again, we repeat the experiments with and without assigning weights to triplets. In the weighted case, increasing $m'$ also improves the results. However, the improvement gained by increasing $m'$ is not as much as increasing $m$. This is due to the fact that many triplets are required to capture the local structure around every point. However, the global structure of the data can be explained by a smaller number of triplets.

Finally, Figure 8 illustrate the result of adding randomly generated triplets. We fix $m = 5$, $m' = 10$ and vary the number of random triplets per point $s$. We repeat the experiments with and without assigning weights. As can be seen, in the weighted triplet setting, adding randomly generated triplets introduces more global structure into the embedding. However, this does not hold for unweighted triplets.

In conclusion, assigning weights to the triplets is a crucial piece in obtaining good low-dimensional embeddings of the data. Without adding weights, we may need much larger number of triplets to achieve similar performance as the weighted case. Additionally, exploring larger number of nearest neighbors provide more information about the local structure of the data whereas the global structure is explained by a fewer number of distant points. Additionally, a small number of randomly generated triplets improves the global structure of the embedding.

## D  ADDITIONAL EXPERIMENTS SHOWING ON MANIFOLD DATASETS

In this section, we show experimental results on datasets with known underlying low-dimensional manifolds. Our datasets include: 1) 3-D Sphere dataset[22], which contains 1000 points uniformly sampled from a unit sphere, 2) Swissroll dataset[23], which is a 2-D rectangle (1500 points) curved rolled into a 3-D shape, and 3) ISOMAP Faces dataset[24], which consists of 698 synthetically generated $128 \times 128$ face images in different pose and lighting. We show the runtime of the t-SNE, LargeVis, and TriMap on these datasets in Table 2. One major point we noticed during our experiments is that LargeVis performs very slow on: 1) small datasets (see Table 2), and 2) datasets with large number of dimensions (Table 1, TV News). The former could be an implementation issue while the latter is due the random projection step for nearest neighbor search. Note that we perform PCA as a step in our algorithm to accelerate the nearest neighbor search (also used in the original t-SNE implementation). The error induced by the PCA step is negligible compared to the error of mapping from high-dimension to 2-D or 3-D. We can also accelerate LargeVis on large dimensional datasets by performing the PCA step (although it is not part of the original algorithm). However, the performance on small datasets still remains slow.

| Dataset (size, dimension) | t-SNE | LargeVis | TriMap |
|---|---|---|---|
| Sphere (1000, 3) | **0-00:00:07** | 0-00:06:00 | 0-00:00:09 |
| Swissroll (1500, 3) | **0-00:00:09** | 0-00:06:12 | 0-00:00:14 |
| ISOMAP Faces (698, 4096) | **0-00:00:08** | 0-00:06:28 | **0-00:00:08** |

Table 2: Runtime of the methods in `d-hh:mm:ss` format.

Figure 9 shows the results of t-SNE, LargeVis, and our method, TriMap, on the Sphere and the Swissroll datasets. As can be seen, TriMap successfully preserve the continuity of the underlying manifold in both datasets. On Sphere dataset, the TriMap visualization corresponds to pulling the 3-D sphere along the diagonal and collapsing it on 2-D. On Swissroll dataset, note that TriMap preserves the curvature of the original dataset; the end parts of the rectangle (colored in dark blue and yellow) are actually closer in the original 3-D space (compare this to the result of LargeVis). In Figure 10, we illustrate the results of t-SNE, LargeVis, PCA, PHATE, UMAP, and our method, TriMap, on the ISOMAP faces daatset. PHATE, PCA, and TriMap uncover the curvature on two end-points of the underlying manifold where the sets of darker images having different poses are located. These darker images (having different poses) are intuitively closer to each other in the high-dimensional space than their brighter counterparts.

---

[22] https://research.cs.aalto.fi/pml/software/dredviz/
[23] https://www.mathworks.com/products/demos/machine-learning/swiss_roll/swiss_roll.html
[24] http://web.mit.edu/cocosci/isomap/datasets.html

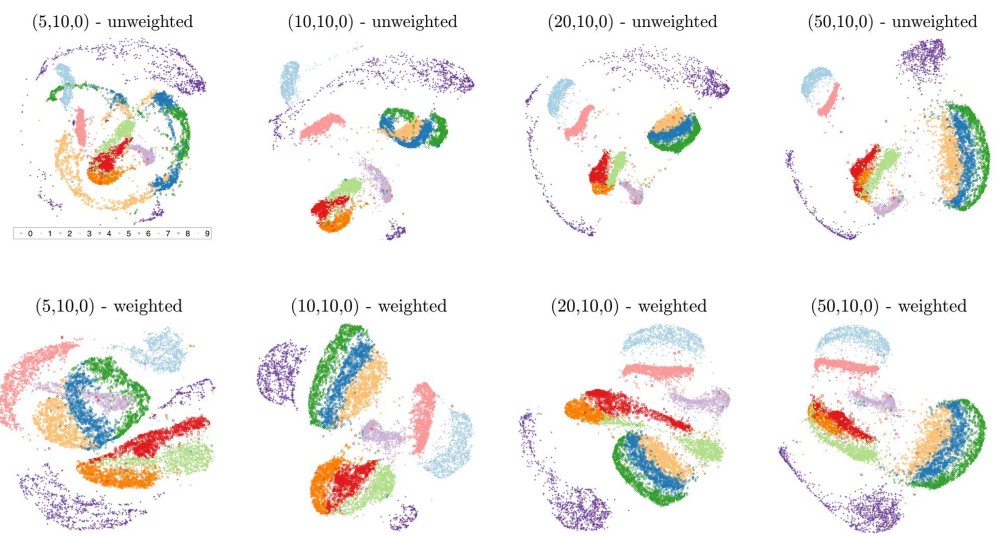

Figure 6: The effect of changing the number of nearest neighbors $m$ using unweighted triplets (top row) and weighted triplets (bottom row). The values $(m, m', s)$ are denoted on top of each figure. Note that increasing $m$ consistently improves the results when using weighted triplets, while the improvement for unweighted triplets is less apparent.

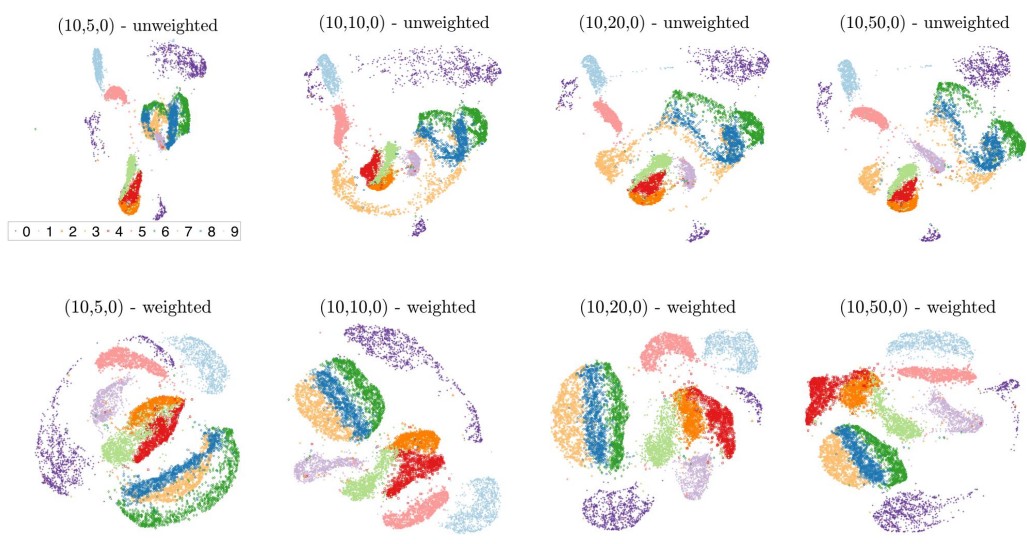

Figure 7: The effect of changing the number of distant points $m'$ using unweighted triplets (top row) and weighted triplets (bottom row). The values $(m, m', s)$ are denoted on top of each figure. Note that increasing $m'$ also improves the results of weighted triplets. However, the local properties are not as much affected by the distant points. Again, the improvements are less evident using the unweighted triplets.

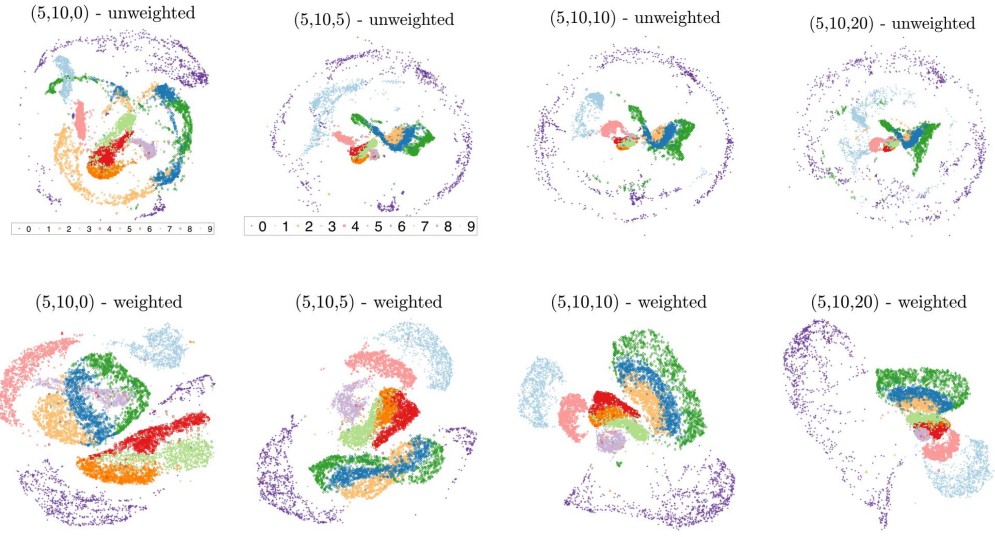

Figure 8: The effect of changing the number of random triplets per point $s$ using unweighted triplets (top row) and weighted triplets (bottom row). The values $(m, m', s)$ are denoted on top of each figure. Note that increasing $s$ improves the global structure, obtained using the weighted triplets. The unweighted triplets do not benefit from randomly generated triplets.

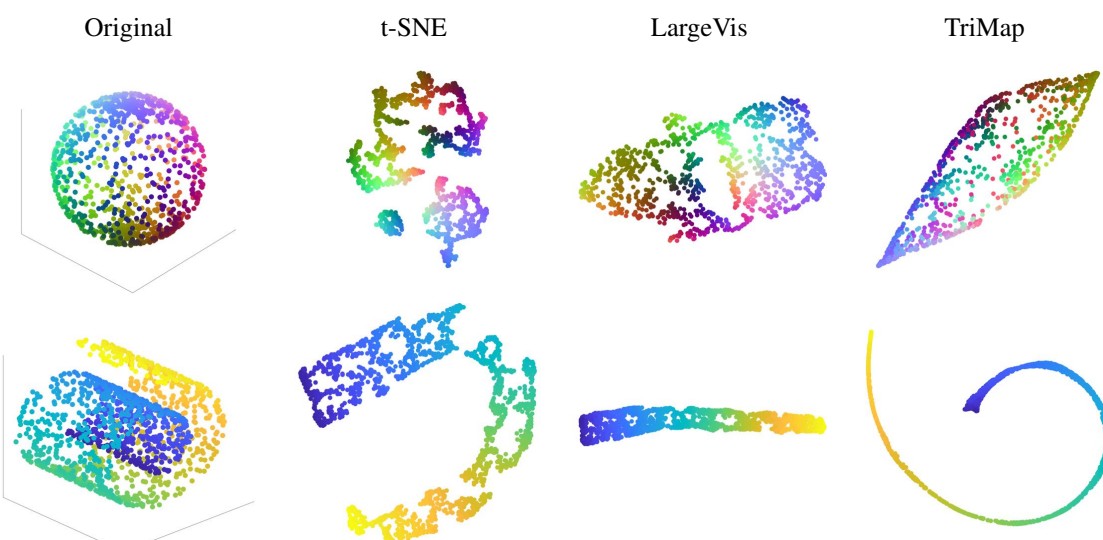

Figure 9: Results of different method on the Sphere dataset (top) and Swissroll dataset (bottom). The visualization of the Sphere dataset by TriMap corresponds to stretching the sphere along the diagonal and spreading the points on 2-D. Note that the visualization by TriMap preserves the continuity of the underlying manifold more than t-SNE and LargeVis. On Swissroll dataset, TriMap preserves the global structure, e.g. relative distances between the end-points of the underlying manifold, better than the other methods.

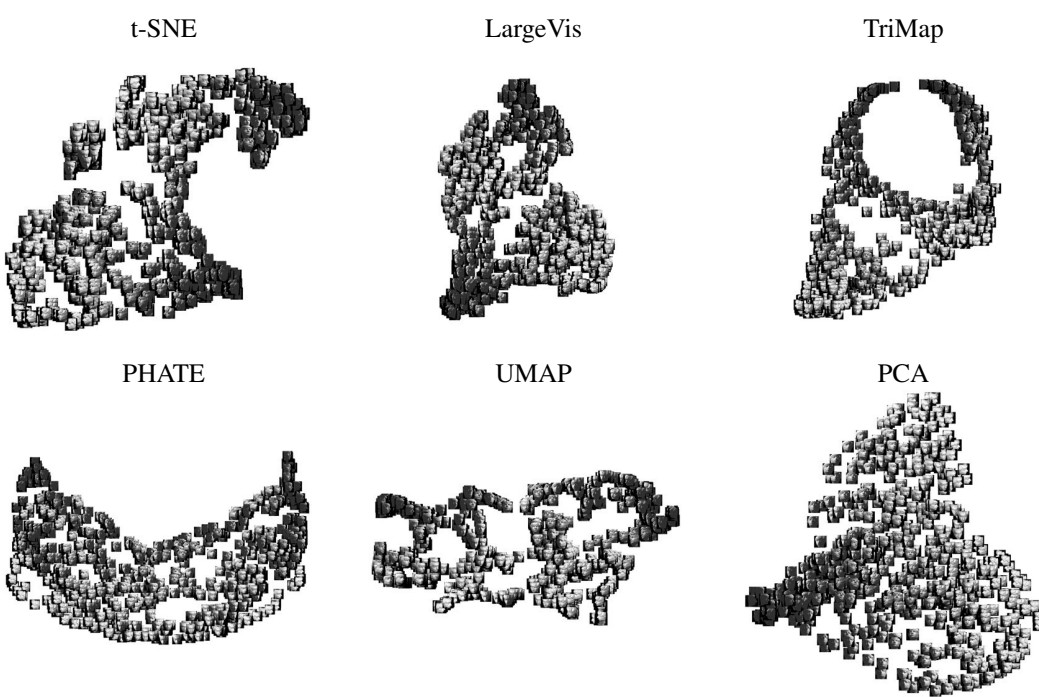

Figure 10: Results of different method on the ISOMAP Faces dataset. PHATE, PCA, and TriMap uncover the curvature on two end-points of the underlying manifold where the sets of darker images having different poses are located. These darker images (having different poses) are intuitively closer to each other in the high-dimensional space than their brighter counterparts.

