# OpenReview forum: "A More Globally Accurate Dimensionality Reduction Method Using Triplets"
_ICLR.cc/2019/Conference_

### Official Review · AnonReviewer1 · 2018-10-26
**A More Globally Accurate Dimensionality Reduction Method Using Triplets**

**Rating:** 6
**Confidence:** 3

**Review:**

Authors propose a new method called TriMap, which captures higher orders of structure with triplet information, and minimize a roust loss function for satisfying the chosen triplets.

The proposed method is motivated by the misleading selection approach for a dimensionality reduction method using local measurements. And then, authors resort to an evaluation based on visual clues based on a number of transformations. Authors then claim that any DR method preserving the global structure of the data should be able to handle these transformations.  An example on MNIST data illustrate these properties, but it is still not clear what are the visual clues as the criterion to select a good DR method and what are the global structures.

Authors discussed the results in Figure 4 for six real-world datasets, but there is no convincing evidence from the corresponding domains or reference researches for the support of the global structure in the learned embedding space.  It will be good to add some convincing evidences for the conclusion.

As the method highly depends on the subset of sampled triplets, it is interesting to see how the global structure changes if a different set of triplets is used.  In addition, it is unclear why sampled triplets can achieve a global structure of data instead of pairwise relations. From the experiments, triplets are also sampled according to the pairwise nearest neighbor graph.

---

### Official Review · AnonReviewer3 · 2018-11-03
**TripMap needs more comparison and validation**

**Rating:** 5
**Confidence:** 4

**Review:**

In this paper, the authors present a novel dimensionality reduction method named TriMap. TriMap attempts to improve upon the widely-adopted t-SNE algorithm by incorporating global distances through the use of triplets, rather than pairwise comparisons. The authors compare to t-SNE, as well as a newer method called LargeVis which also claims to impose a global distances metric. The authors show that their method is more robust to the addition or removal of clusters and outliers and provides a more meaningful global distance relative to the methods against which they compare.

Technical Quality
The authors’ method is clear and well described and addresses a poignant issue in dimensionality reduction. However, the authors fail to compare their method to a number of relevant dimensionality reduction algorithms which also claim to provide solutions with globally meaningful distances. Such methods include force-directed graph drawing (Fruchterman & Reingold, 1991), diffusion maps (Coifman & Lafon, 2006) and PHATE (Moon et al., 2017).

Additionally, the handling of outliers is a concern. While the authors claim that the retention of outliers as disconnected from the manifold is a desirable quality of their technique, the presence of many outliers in a dataset (for example, in the Tabula Muris and lyrics datasets) has the potential to mask the interesting portion of the dimensionality reduction. It may be worth commenting on the desirability to identify and remove outliers, and the provision of such a technique in the software upon its release.

Finally, the runtime comparison is of concern. It is common to perform most DR methods on high-dimensional PCA representation of the data, particularly in single-cell genomics (e.g. the Tabula Muris dataset in Part 3.) In this context, both UMAP and PHATE successfully embed the Tabula Muris dataset in less than the reported TriMap time (3.5 and 5 minutes respectively, compared to 15 minutes reported for TriMap.)

Novelty
The authors’ method appears to provide improved results over the compared alternatives, however, it is worth noting that triplet-based embedding is not novel in its own right (van der Maaten & Weinberger, 2012), though one could argue novelty is warranted here due to claimed substantial improvements of results. In this case, the authors should include a comparison to competing triplet-based methods, at least in the appendix.

Potential impact
The authors’ method has the potential to be used widely across many fields, as a direct replacement for t-SNE. Its adoption is contingent on compelling evidence that it produces results substantially better than UMAP (which is currently heralded as an upcoming replacement for t-SNE in some fields) and other competing methods. The authors may find it worthwhile to provide such comparisons, if not in the main body of the paper at least in the appendix.

Clarity
The paper is easy to read and makes its point in a reasonably concise manner. Detailed explanation of experiments v) and vi) could be relegated to the appendix. More precise statement of the authors’ tests in Part 2 could be provided by quantifying the results of the tests in a more precise way; it is not clear what the authors seek to achieve by drawing the dotted lines between clusters in Figure 1a, or by providing AUC values in Figure 1.

Detailed Comments
•	In the definition of Equation 2, it is not until one paragraph later than q_{ij}^{~(t’)} is defined – this is confusing and hard to read.
•	The captions for Figures 1 and 3 would be substantially clearer with more detail on the dataset analyzed and in Figure 1, some discussion of the purpose of each subplot.
•	The Figure 3 caption needs a semicolon or period before introducing the bottom panel.
•	The claim that the authors’ heuristic triplet sampling (nearest-neighbor and random sampling) is sufficient to approximate full triplet sampling should be shown in the appendix.
•	The collaboration network analyzed in Part 3 is naturally a graph; it would make sense to cluster and visualize this using a graph-based clustering, rather than coercing it to Euclidean coordinates.

(Note: after reading the revised manuscript I have changed my recommendation from a 6 to a 5)

References
Coifman, R. R., & Lafon, S. (2006). Diffusion maps. Applied and computational harmonic analysis, 21(1), 5-30. https://doi.org/10.1016/j.acha.2006.04.006
Moon, K. R., van Dijk, D., Wang, Z., Burkhardt, D., Chen, W., van den Elzen, A., ... & Krishnaswamy, S. (2017). Visualizing transitions and structure for high dimensional data exploration. bioRxiv, 120378. https://doi.org/10.1101/120378
Fruchterman, T. M., & Reingold, E. M. (1991). Graph drawing by force‐directed placement. Software: Practice and experience, 21(11), 1129-1164. https://doi.org/10.1002/spe.4380211102
L. van der Maaten and K. Weinberger. Stochastic triplet embedding. In 2012 IEEE International Workshop on Machine Learning for Signal Processing, pp. 1–6, Sept 2012. doi: 10.1109/MLSP.2012.6349720.

---

### Official Review · AnonReviewer2 · 2018-11-04
**Novel loss function but experiments are lacking**

**Rating:** 6
**Confidence:** 5

**Review:**

Motivated by the observation that most of previous dimensionality reduction methods focus on preserving
local pairwise neighboring probabilities and lack in preserving global properties, this paper proposes a
method called TriMap to optimize a loss function preserving similarities among triplets of data points. A large
number of triplets are sampled either based on nearest neighbor calculations or random sampling. Experimental
results on several datasets show that TriMap identifies outliers and preserves global data properties better
than previous approaches based on pairwise data point comparisons.

Major:

The idea in this paper is well motivated and the loss function based on probability ratio is novel. However,
there are some major concerns about method analyses and experimental evaluations,

1. Data embedding based on triplets has been presented in (van der Maaten and Weinberger, 2012). The authors
need to present detailed explanations and formal analysis why the proposed method significantly outperforms the
previous one. A recent dimensionality reduction method compares data points only to data cluster centers (Parametric
t-distributed stochastic exemplar centered embedding, Min et al., 2018), does it preserve global data properties? Does
its trivial combination with standard t-SNE well preserve both local and global data properties?

2. Preserving local pairwise neighborhood structure is often the most important part in high-dimensional data
visualization, because only local similarities can be confidently trusted in a high-dimensional space. Even if preserving
global data properties is important, the very local neighborhood structure should also be preserved. However, the
proposed method TriMap is significantly worse than t-SNE according to AUC under the precision-recall curve.

3. Standard quantitative evaluations based on 1NN error rate and quality scores (van der Maaten & Hinton 2008, Min
et al. 2018) should be added to the experiments. For preserving global data properties, quantitative evaluations on all
the datasets will make the experiments much more convincing.

4. In the abstract, the claim that TriMap scales linearly is inaccurate, the triplet sampling requires nearest neighbor
calculations, which has computational complexity of at least O(nlogn)

5. This paper proposed two variants of triplet sampling, nearest neighbor triplets and random triplets. Detailed experimental
comparisons about them should be provided in the paper.


6. The running time comparisons in Table 1 must be wrong or highly biased with improper hyperparameter setting. Based
on tree accelerations, t-SNE can produce impressive visualization on MNIST-scale datasets within 15 minutes (please
check the experimental details PP. 3235-3238 in van der Maaten, Journal of Machine Learning Research 2014).

7. The authors mentioned partial observation, outliers and subclusters in the global information, but the authors do not specifically define
what the global information should be rigorously, and the paper does not theoretically prove or explain via experiments how the global
information is kept by TriMap.

8. In the experiments, the authors applied PCA before TriMap to reduce the dimensionality while PCA is not applied in tSNE and LargeVis. The authors do not explain why the settings are different in the three methods.

Minor:

9. In the algorithm, the authors show different equations for different t and t’, but are not evaluated in experiments.

(After reading the rebuttal, I raised the rating from 5 to 6.)

---

### Author Response · Authors · 2018-11-17
**Common Concerns**

Thank you for your thorough reviews.
We were able to significantly our experimental evaluation based on your suggestions.

We first address the common concerns raised by reviewers:

*** Comparison to Stochastic Triplet Embedding (van der Maaten and Weinberger, 2012) ***
In Appendix A, we show that STE (also t-STE) is a special case of our method where the parameters are chosen sub-optimally (in the sense that the loss is not robust) and the triplet weights are set to one. We also added results of the DR tests on t-STE in Appendix B. Also the result in Figure 3 with (t = 1, t' = 2) corresponds to t-STE with the addition of triplet weights. Overall, it is evident from the experiments that t-STE is inferior and provides poor results by introducing
factitious outliers. Additionally, t-STE fails to reveal the true outliers (See Figure 5.2.2).

*** Quantitative Measures of Local and Global Performance ***

The main reason for including AUC (which is a local measure of DR performance) is to show that the local measure do NOT reflect the global properties of the embedding. This is discussed in our tests for global accuracy in Section 2 of the paper. For instance, in Figure 1.2.3, PCA clearly separates the two copies of MNIST, but has a very low AUC score. We also included the NN classification accuracy in our results and (after fixing a minor bug in our code) updated the AUC scores. The (AUC, NN-Accuracy) values are show on the bottom of each plot in Figure 1 and 5. Again, NN-accuracy fails to reflect these global properties.

We are not aware of any global measure of DR method that can reflect the properties discussed in our tests. We leave the development of such measures that are tractable for large datasets for future work. We also added DR test results produced by PHATE (Moon et al., 2017) and UMAP (McInnes and Healy, 2018). These two methods also fail at least some of the DR tests. We also performed experiments using Diffusion Map (Coifman and Lafon, 2016). The Diffusion Map method is incredibly slow and we were only able to calculate the results for a subset of 10,000 points from MNIST. Overall, the method has good global properties and is able to detect the artificial outlier and the two copies. However, the quality of the embeddings are much inferior compared to other methods (results available anonymously at: https://goo.gl/bGJqSD). We were not able to perform the tests on Parametric t-distributed Stochastic Exemplar Centered Embedding (Min et al., 2018). The method requires careful implementation of a neural network and this was not feasible to try in the given timeframe. However, the method is heavily motivated by the pt-SNE (van der Maaten, 2009) method and thus, we conjecture that it also may not be able reflect the global properties. Overall, we are not aware of any competitor method (with comparable runtime and quality of embedding) that can reflect all the global properties discussed in our paper.

We also added experimental results on three datasets with underlying low-dimensional manifolds, namely, 3-D Sphere, Swissroll, and ISOMAP Faces (Figure 9 and 10). TriMap again provides globally accurate results while preserving the continuity of the underlying manifolds. Thus, although the local measures for TriMap are not as high as t-SNE, is still provides locally accurate results.

---

### Author Response · Authors · 2018-11-17
**Common Concerns (Continued)**

*** Runtime and Computational Complexity ***

Thank you for noticing the error in the runtime given in Table 1. We corrected the runtime on Fashion MNIST dataset. Note that we used the sklearn implementation of t-SNE on a 2.6 GHz Intel Core i5 machine with 16 GB memory. We used the same machine for running LargeVis and TriMap and we are certain that the results are accurate. One major point we noticed during our experiments was that LargeVis performs very slow on: 1) small datasets (see Table 2), and 2) datasets with large number of dimensions (Table 1, TV News). The former could be an implementation issue while the latter is due the random projection step for nearest neighbor search. Note that we perform PCA as a step in our algorithm to accelerate the nearest neighbor search (also used in t-SNE; please see the original paper and the sklearn implementation details). The error induced by the PCA step is negligible compared to the error of mapping from high-dimension to 2-D or 3-D. We can also accelerate LargeVis on large dimensional datasets by performing the PCA step (although it is not part of the original algorithm). However, the performance on small datasets still remains slow.

Thank you also for correcting our statement about our computational complexity. The nearest neighbor search is indeed the bottleneck of most of the algorithms (t-SNE, LargeVis, UMAP, TriMap, etc.). In some sense, all these algorithms have at least O(n log n) complexity due to the NN seach. However, notice that after the nearest neighbor search step (which is done once), (Barnes-Hut) t-SNE still has O(n log n) complexity while our method scales linearly. We have already corrected the complexity discussion in the current version. We are actively working on improving the runtime of TriMap. Our goal is to achieve a runtime as fast as UMAP.

*** Effect of Weights and Different Triplets ***

We added extra experimental results to show the effect of adding weights to the triplets as well as the effect of different types of triplets (nearest neighbors triplets vs randomly generated triplets). The results are given in Figure 6, 7, and 8. In conclusion, assigning weights to the triplets is a crucial piece in obtaining good low-dimensional embeddings of the data. Without adding weights, we may need much larger number of triplets to achieve similar performance as the weighted case. Additionally we show that using a larger number of nearest neighbors for forming triplets provides more information about the local structure of the data  whereas the global structure is explained by a fewer number
of distant points. Also a small number of randomly generated triplets improves the global structure of the embedding.

---

### Author Response · Authors · 2018-11-17
**Detailed Comments**

Reviewer 1:

Thank you for raising the concerns about sampling triplets. We hope that additional experimental results given in Appendix C addresses some of your questions. In addition, although the triplets are formed using the pairwise distances, they simultaneously take into account the relative similarities (or distances) of three points. Therefore they are somehow more informative than simple pairwise constraints. This has been discussed before in the semi-supervised clustering context: e.g. in "A kernel-learning approach to semi-supervised clustering with relative distance comparisons" by Amid et al. 2015. They showed experimentally that a smaller number of triplet constraints improves the performance of clustering significantly more than pairwise constraints.


Reviewer 2:

Although the local properties of the low-dimensional embeddings are highly valuable, we claim that the global properties are sometimes much more important in certain scenarios. For instance, in medical diagnosis, the relative closeness of the clusters of points or detection of bad outliers (e.g. cancerous vs. non-cancerous cells) are highly critical. Our paper brings out this issue and provides a solution for improving the global accuracy. We are hoping that our paper will initiate new research on DR methods that focus on global properties.

Reviewer 3:

Thank you for your suggestion on detecting and removing the outliers. We are planning to add the "zoom feature" to our official release of the code. We would like to enable enhancing the visualization of sub-regions of the embedding by simply zooming into the region and re-evaluating the algorithm on the subset of the points. Since TriMap is rather insensitive to removing portions of data (experimental evidence in Figure 1.2.1a and 1.2.1b), this feature would fit perfectly with method.

The dotted line between the clusters of '0's and '6's is drawn to mainly emphasize that the relative distance between these two clusters (and the rest) before and after removing the odd digits remains almost the same for our method but change significantly when using LargeVis (Figure 5.\star and Figure 2.1.b). This dotted line was unnecessary for Figure 1.2.1a and therefore, we removed it in the updated version.

---

> ### Comment · AnonReviewer2 · 2018-11-24
> **Quantitative definition about global properties and detailed comparisons to t-SNE missing**
>
> Thanks for the authors' detailed explanations and experiments addressing the raised concerns. After reading the authors' rebuttal, I still have two major concerns that I believe to be highly important:
>
> 1. TriMap produces good visualization without sacrificing global properties too much. However, no formal quantitative definitions or evaluations are provided, which is the key to support the claimed advantages of the proposed method. Without these quantitative or theoretical analyses, users will be very reluctant to choose such a method lying between other methods. If the users' target is to identify outliers preserving global properties, they will choose other embedding methods that directly focusing on anomaly detection (for e.g., recent methods based on deep autoencoders).
>
> 2. Comparing to the default implementation of t-SNE in sklearn is highly biased. Before applying t-SNE with tree accelerations, PCA should also be applied to the original data to obtain reasonably low-dimensional (e.g., 100-d) input data to t-SNE, through which the running time of tree accelerated t-SNE will be significantly reduced.  The comparisons to tree accelerated t-SNE in this paper is biased.
>
> Therefore, item 3, 6, and 7 in my original review should be seriously addressed.
>
> After reading the rebuttal, I revised my rating from 5 to 6.

---

> > ### Author Response · Authors · 2018-11-24
> > **Further Discussion**
> >
> > Thank you for acknowledging our response.
> >
> > 1) Your concern about the global measures of DR performance is absolutely valid. We are not aware of any global measure of DR method that can reflect the properties discussed in our tests. However, we are actively working on developing such measures that are tractable for large datasets. For instance, a direct generalization of precision-recall (which are local measures of DR) to global measures would be to consider the "farthest-away remoteness" of each point. That is, for each point i, (instead of nearest-neighbors) consider the farthest-away points from i in the high-dimensional space and the low-dimensional embedding. The ratios (true farthest points recovered/total points recovered) and (true farthest points recovered/total true farthest points) would be direct generalizations of precision and recall, respectively. However, while calculating nearest-neighbors is relatively cheap, finding farthest-away points in high-dimension is extremely inefficient and cannot be used for datasets of our scale. We leave the development of methods for (approximately) calculating such measures on large datasets for future work.
> >
> > 2) We checked different implementations of t-SNE. There seems to be some inconsistency about applying PCA in the official implementation of t-SNE (https://lvdmaaten.github.io/tsne/) and the sklearn implementation (https://scikit-learn.org/stable/modules/generated/sklearn.manifold.TSNE.html). We will update the runtime of t-SNE and LargeVis in Table 1 and 2 and include the runtime before and after applying PCA down to 100 dimensions.

---

### Author Response · Authors · 2018-11-27
**A Gentle Reminder**

Thanks again to all reviewers.
This is just a gentle reminder (to Reviewer 1 and 3) to possibly give us feedback on our new additions.

---

### Comment · AnonReviewer3 · 2018-12-03
**Insufficiently complete revision**

The proposed method appears promising and seems to work well on the examples shown in the paper. However, the authors' additional comparisons competing methods are very cursory and do not substantially demonstrate Trimap's superiority to the existing suite of dimensionality reduction methods currently available. The authors do not provide sufficient discussion to justify the formulation of Trimap or to provide a theoretical basis for its performance over other methods.

While I believe the algorithm is a valuable contribution, the presentation of the algorithm in this manuscript is not sufficient for its publication. The authors would do well to revise the manuscript to give a clearer justification, including a conclusive suite of benchmark datasets. Additionally, further evidence for the value of Trimap could be provided in the analysis of its scalability to tens or hundreds of dimensions for use as a general dimensionality reduction methods, as is done with many methods like PCA and diffusion maps, but is not possible with t-SNE. This would make Trimap useful for applications such as clustering, rather than only for visualization.

In light of this cursory revision of the manuscript, I have updated my recommendation from a 6 to a 5.

---

> ### Author Response · Authors · 2018-12-03
> **Comments are vague and unclear**
>
> Please note that we carefully addressed all the concerns raised in your initial review. In the revised version, we showed that other DR methods (PHATE, UMAP, and STE) fail at least some of the global tests discussed in our paper. Note that we did not provide thorough comparisons with PHATE and UMAP because these two methods are unpublished work! We believe we have provided a clear problem formulation, a performance comparison with two state-of-the-art methods, t-SNE and LargeVis, and proof for scalability to tens or hundreds of dimensions. Furthermore, we added evidence that each piece in our formulation (loss transformation, triplet weighting, nearest-neighbor triplets, and random triplets) are crucial for our method to work. The only concern not addressed in our work in a "global measure of performance" for DR methods, which actually has not been developed yet and we are not aware of any measure that can quantitatively reflect the performance on our DR tests.
>
> While we appreciate your feedback very much, your final comments seem rather vague and subjective. We need more concrete and clear comments to be able to improve our paper. We appreciate it very much if you provide a "detailed reasoning" why you would think the revised version deserves a lower score than the initial submission. While we respect your decision, lowering the score after significantly improving the comparisons and adding clarifications to the revised version while addressing all your concerns in the initial review is rather unfair.

---

### Meta-Review · Area_Chair1 · 2018-12-14
**A new take on dimensionality reduction which deserves a more guided experimental section**

**Confidence:** 4
**Recommendation:** Reject

**Metareview:**

Dear authors,

The reviewers all appreciated your goal of improving dimensionality reduction techniques. This is a field which does not enjoy the popularity it once did but remains nonetheless important.

They also appreciated the novel loss and the use of triplets.to get the global structure.

However, the paper lacks some guidance. In particular, it oscillates between showing qualitative results (robustness to outliers, "nice" visualizations) and quantitative ones (running time, classification performance). I agree with the reviewers that the quantitative ones should have used the same preprocessing for t-SNE and TriMap (either PCA or no PCA), regardless of the current implementation in software tools.

Given that the quantitative results are not that impressive, may I suggest focusing on the qualitative ones for a resubmission? The robustness of the emeddings to the addition or removal of a few points is definitely interesting and worth further investigation, optionally with a corresponding metric.